# Mortality as the primary constraint to enhancing nutritional and financial gains from poultry: A multi-year longitudinal study of smallholder farmers in western Kenya

**Elkanah Otiang**[1,2,3], **Zoë A. Campbell** [4,5], **Samuel M. Thumbi**[1,2,3,4], **Lucy W. Njagi**[1], **Philip N. Nyaga**[1], **Guy H. Palmer** [1,3,4] *

**1** University of Nairobi, Nairobi, Kenya, **2** Kenya Medical Research Centre, Kisumu, Kenya, **3** Washington State University Global Health-Kenya, Nairobi, Kenya, **4** Paul G. Allen School for Global Animal Health, Washington State University, Pullman, Washington, United States of America, **5** International Livestock Research Institute, Nairobi, Kenya

\* gpalmer@wsu.edu

**Data Availability Statement:** The full data set is provided in Open Science Framework at https://osf.io/9u4fn/.

## Abstract

### Background

Chickens are a widely held economic and nutritional asset in rural Africa and are frequently managed by women. Despite potential benefits of larger flock sizes, the average number of chickens kept at the household level is reported to be low. Whether this reflects decision-making to maximize benefits per unit labor by voluntary reduction of chicken numbers by consumption or sale versus involuntary losses due to mortality is a significant gap in knowledge relevant to improving smallholder household welfare.

### Methods

In a 4-year longitudinal study of 1,908 smallholder households in rural western Kenya, the number of chickens owned by quarterly census at each household was determined. Households reported gains and losses of chicken over the immediate previous quarter. Gains were classified as on-farm or off-farm; losses were classified as voluntary (sales, gifts, consumption) or involuntary (mortality, unclassified loss).

### Results

The mean number of chickens owned over the 16 quarters was 10, consistent with prior cross-sectional data. Involuntary losses represented 70% of total off-take, while voluntary off-take represented the remaining 30%. Mortality composed 60% of total reported off-take and accounted for most of the involuntary losses. Household consumption, sales, and gifts represented 18%, 9%, and 3% of off-take, respectively.

**Funding:** Funding was provided by the Paul G. Allen School for Global Animal Health Washington State University. There is no specific grant number. https://globalhealth.wsu.edu Neither the late Paul G. Allen or his Foundation had any role in study design, data collection and analysis, decision to publish, or preparation of the manuscript.

**Competing interests:** The authors have declared that no competing interests exist.

## Conclusion

The overwhelming majority of off-take can be classified as involuntary off-take, principally due to mortality, that does not reflect the owner's decision to maximize value through nutritional gain, income, or social capital. This strongly suggests that there is substantial opportunity to enhance the value of chickens as an asset, both nutritional and income generating, for smallholder households living at poverty level. Our findings suggest that programs emphasizing community level poultry vaccination and feed supplementation are much more likely to be effective than those solely focused on providing chickens.

## Introduction

Improving small scale poultry production has been targeted as a pathway to improve nutritional and economic gains for rural African families affected by high rates of childhood stunting and poverty. Chickens kept by smallholder farmers serve as a source of food rich in protein and micronutrients (eggs and meat), household income through sales, investment, and savings [1–4]. In addition, chickens have an important cultural role in societal and familial life [2]. Chickens are frequently managed by women, providing opportunity to enhance gender equality in resource management [5–8]. This, in turn, may result in healthier households as women are more likely to prioritize spending on health care, nutrition, and education [8]. The commitment of organizations such as the Bill & Melinda Gates Foundation, Heifer International, the U.N. Food and Agriculture Organization, and others in encouraging chicken-keeping and improving smallholder poultry productivity in sub-Saharan Africa illustrates the potential for achieving broader health and development goals [3,9].

Despite these known and potential benefits, smallholder flock sizes remain small. In a cross-sectional survey of smallholder households in Tanzania, mean ownership was 11 chickens in a 2007 study [10] and 13 chickens as reported in two independent studies from 2010 and 2018 [3,11]. Similarly, in Kenya, smallholder flock sizes ranged between 10 and 20 chickens [12–15]. Whether this consistent small flock size reflects deliberate management choices or involuntary losses is a significant gap in knowledge relevant to ongoing programs encouraging smallholder poultry ownership. Although smallholder flocks typically scavenge during the day for a significant proportion of their feed and are often housed only at night, even this minimal management has an opportunity cost for the household, with the greatest cost to women. Production priorities are likely to differ between men and women when production increases do not allow women to benefit in proportion to their labor contributions [6]. Increasing the number of chickens beyond the current flock sizes may impact time available for other activities that fall predominantly on women such as gathering fuel and water, cooking, and child-care. A large proportion of voluntary off-take suggests a household is actively managing flock size and optimizing benefits per available unit of labor. Examples of voluntary off-take include sales, giving chickens as gifts, and household consumption of chickens. In contrast, when a large proportion of loss is involuntary, such as death from disease, starvation, or predation, it suggests the household is missing out on potential nutritional and economic benefits per unit labor.

We address this question of why flock sizes remain small using a longitudinal survey between 2013 and 2017 of approximately 1,900 households in ten villages in rural western Kenya. A 2013–2014 study reported that 80% of these households depend on mixed-crop and livestock agriculture and 88% kept chickens [15]. In the current longitudinal study, a census of chicken ownership was conducted quarterly for each household, and included recall of

increases due to purchases, gifts, and on-farm hatching as well as decreases due to sale, consumption, gifts, death, and unknown losses such as predation. Here, we present the findings of the study, and discuss the results in the context of understanding the factors that drive small flock sizes in smallholder production systems with the goal of enhancing benefits of smallholder chicken ownership.

## Materials and methods

### Ethical approval

The research was approved by the Ethical and Animal Care and Use Committees (SSC Protocol no. 2250) of the Kenya Medical Research Institute to actively conduct animal disease surveillance and economic impact of domestic animal ownership, morbidity, and mortality.

### Study population

A total of 1,908 rural households with a total population of approximately 6,400 individuals in Rarieda sub-county, Siaya County, Kenya were enrolled and followed longitudinally over 16 quarterly visits starting in February, 2013. The mean age of household head was 53 years with a third of households having at least a child <3 years of age and 42% with a child < 5 years old (Table 1). A cross-sectional study in 2015 found that 23.5% of children <5 years in these households were physically stunted, underscoring the importance of nutrition in this population [16]. Households practice small-scale mixed livestock and crop agriculture for livelihoods. At study initiation, 93% of households reported owning at least one species of livestock, and 88% reported owning chickens. The median number of ruminant livestock was nine per household, which included cattle (55% of households), goats (41%), and sheep (19%). Primary crops were maize (produced by 91% of households), beans (35%), potatoes (8%), sorghum (5%), and cassava (4%) [15]. All households were included in the study even if they did not own chickens at study initiation as chicken ownership may fluctuate by season or be affected by external factors such as periodic off-farm employment opportunities.

**Table 1. Demographic characteristics of the study population.**

| Age of Primary Respondent | 53.3±17.1 years |
|---|---|
| Gender | |
| Male | 50.1% |
| Female | 49.9% |
| Education Level | |
| No formal education | 14.5% |
| Primary education | 65.4% |
| Secondary education | 16.6% |
| Tertiary education | 3.6% |
| Primary Occupation | |
| Employed full-time on farm | 56.6% |
| Employed part time on farm | 9.0% |
| Self-employed off farm | 16.3% |
| Salaried off farm | 5.3% |
| Other | 12.8% |
| Households with ≥ 1 child | |
| Under 3 years | 33% |
| Under 5 years | 42.3% |
| Under 10 years | 56.3% |

### Data collection

Data on the number of chickens owned on the interview day and recall of gains and losses over the previous three months were collected from each participating household quarterly. These data were collected by community enumerators in the local language, entered onto an electronic data capture tool, downloaded, and stored in a Microsoft Access database®. Increases in chickens over the last quarter were reported by an adult household member and categorized as hatched on premises, purchased, or acquired as a gift. Decreases in chickens were reported as sold, died, lost due to undetermined cause, or given as gifts. For sales, the price per chicken was reported. For income (earnings and sales) and expenses, Kenya shillings were converted to U.S. dollars using 2013 year's average as an exchange rate reference (1 USD = 85.52 Shillings) and discounted at an annual rate of 3.5% [17]. All datasets underwent validation and consistency checks to identify and resolve errors. The full data set is provided in Open Science Framework at https://osf.io/9u4fn/.

### Data analysis

Analysis of flock size and increases and decreases over time were analyzed using STATA (Stata, 2013). Decreases in chicken numbers were categorized as voluntary (consumption, sale, or gifts) or involuntary (death or unclassified loss such as predation or theft).

## Results and discussion

### Chicken ownership over time

The overwhelming majority of households (1,805/1,908; 94.6%) reported owning at least one chicken during at least one of the 16 quarterly visits. This is in broad agreement with prior data from this community that reported 88% of households owning chickens [15]. The average flock size determined by quarterly on-farm census was approximately 10 chickens and was relatively constant over multiple years (Table 2). This longitudinal data supports previous cross-sectional data of chicken ownership in rural east Africa [3,10–15].

### Sources of increases in chicken ownership

Households reported mean gains of 6.38±8.26 chickens per quarter (Table 3). Chicks hatched at the household represented the overwhelming majority of reported increases (97%), with purchases and received gifts representing only 2% and 1%, respectively (Fig 1). The relatively constant flock size suggests that gains through hatching are balanced by off-take.

### Sources of decreases in chicken ownership

Households reported mean off-take of 5.35±7.68 chickens per quarter (Table 4). Involuntary losses represented 70% of total off-take, while voluntary off-take represented the remaining 30%. Mortality composed 60% of total reported off-take and accounted for most of the involuntary losses (Fig 2). Household consumption, sales, and gifts represented 18%, 9%, and 3% of off-take, respectively. Household visits that took place within rainy versus dry seasons were analyzed for an impact of seasonality: there was no significant effect on either gains and decreases in chicken numbers by season nor a significant effect on the source of gains or decreases. In aggregate, the overwhelming majority of off-take can be classified as involuntary, off-take that does not reflect the owner's decision to maximize value through nutritional gain, income, or social capital.

**Table 2. Longitudinal census of household chicken ownership in Rarieda, Kenya.**

| Visit Quarter | Mean # of Chickens/ Household | 95% confidence interval |
|:---:|:---:|:---:|
| 1 | 11.65 | 11.14–12.15 |
| 2 | 9.78 | 9.36–10.21 |
| 3 | 10.45 | 9.97–10.92 |
| 4 | 9.98 | 9.50–10.46 |
| 5 | 8.31 | 7.90–8.72 |
| 6 | 11.10 | 10.59–11.61 |
| 7 | 9.92 | 9.46–10.39 |
| 8 | 9.70 | 9.28–10.12 |
| 9 | 10.65 | 10.19–11.11 |
| 10 | 11.26 | 10.80–11.71 |
| 11 | 11.45 | 10.99–11.91 |
| 12 | 9.34 | 8.94–9.73 |
| 13 | 8.10 | 7.74–8.47 |
| 14 | 7.44 | 7.12–7.77 |
| 15 | 9.51 | 9.11–9.91 |
| 16 | 9.94 | 9.54–10.35 |
| **Cumulative** | **9.91** | **9.48–10.35** |

## Evidence of bias in self-reported gains and losses

There is a slight but notable inconsistency in self-reported quarterly gains and losses as compared to the census data (Fig 3). The recalled gains are on average higher than recalled losses, suggesting the numbers of chickens should marginally increase over time. This is not

**Table 3. Longitudinal household reporting of chicken gains in Rarieda, Kenya.**

| Visit Quarter | Total Gains Mean (95% C. I.) | On-farm Gains[1] Mean (95% C. I.) | Off-farm Gains[2] Mean (95% C. I.) |
|:---:|:---:|:---:|:---:|
| 1 | 7.51 (7.14–7.89) | 7. 34 (6.97–7.71) | 0.17 (0.13–0.21) |
| 2 | 5.71 (5.41–6.020 | 5.57 (5.27–5.87) | 0.15 (0.12–0.18) |
| 3 | 7.23 (6.84–7.63) | 7.05 (6.65–7.45) | 0.18 (0.14–0.23) |
| 4 | 7.67 (7.24–8.10) | 7.41 (7.00–7.82) | 0.26 (0.16–0.36) |
| 5 | 5.85 (5.44–6.25) | 5.61 (5.24–5.98) | 0.24 (0.13–0.34) |
| 6 | 6.89 (6.46–7.33) | 6.66 (6.24–7.08) | 0.23 (0.11–0.36) |
| 7 | 6.57 (6.15–7.00) | 6.34 (5.93–6.76) | 0.23 (0.12–0.34) |
| 8 | 6.01 (5.65–6.37) | 5.83 (5.47–6.19) | 0.18 (0.14–0.22) |
| 9 | 6.33 (5.97–6.70) | 6.18 (5.83–6.53) | 0.16 (0.05–0.26) |
| 10 | 7.38 (6.98–7.80) | 7.14 (6.73–7.54) | 0.24 (0.16–0.32) |
| 11 | 7.62 (7.22–8.02) | 7.50 (7.10–7.90) | 0.12 (0.09–0.14) |
| 12 | 6.44 (6.05–6.84) | 6.32 (5.93–6.71) | 0.13 (0.10–0.15) |
| 13 | 4.85 (4.53–5.17) | 4.68 (4.38–4.98) | 0.17 (0.07–0.28) |
| 14 | 4.10 (3.84–4.36) | 3.97 (3.71–4.22) | 0.14 (0.10–0.17) |
| 15 | 5.90 (5.57–6.23) | 5.79 (5.46–6.12) | 0.11 (0.09–0.14) |
| 16 | 5.95 (6.01–6.75) | 5.81 (5.50–6.13) | 0.14 (0.10–0.17) |
| **Cumulative** | **6.38 (6.01–6.75)** | **6.20 (5.84–6.56)** | **0.18 (0.11–0.24)** |

[1]Hatching on premises

[2]Purchases and received gifts to the household

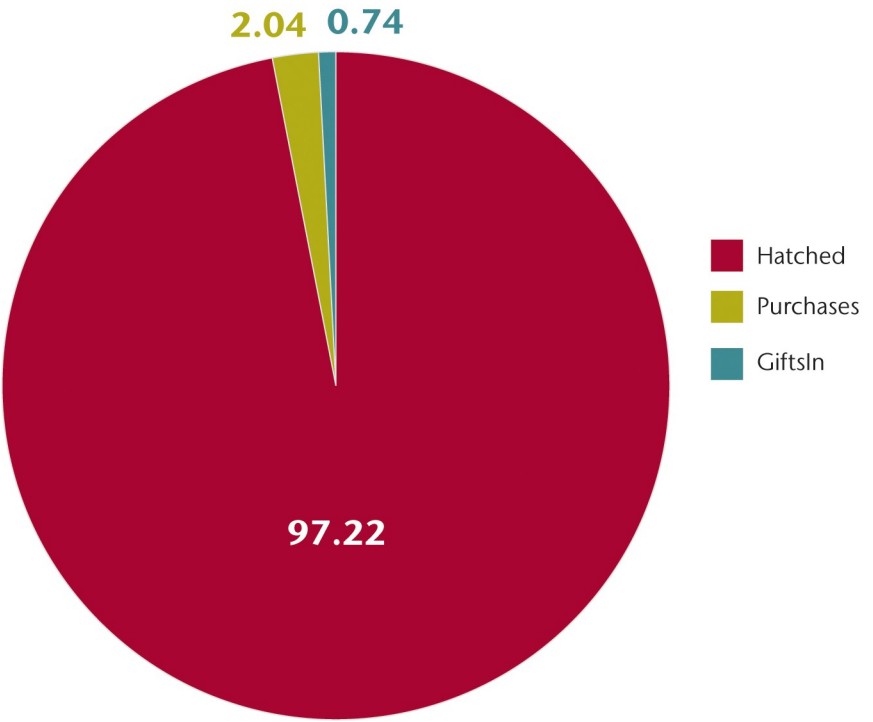

**Fig 1. Source of reported gains in chicken ownership (percentages).**

consistent with the census, which shows flock size remaining stable over time. A notable difference is that the self-reported gains and losses require recall over the prior quarter while the census is based on an actual count. Two possible explanations for the discrepancy are recall bias and social desirability bias. Recall bias, or systematic errors in remembering past events or

**Table 4. Longitudinal household reporting of chicken off-take in Rarieda, Kenya.**

| Visit Quarter | Total Off-take Mean (95% C.I.) | Voluntary Off-take Mean (95% C.I.) | Involuntary Losses Mean (95% C.I.) |
|---|---|---|---|
| 1 | 6.95 (6.49–7.40) | 1.97 (1.82–2.12) | 4.97 (4.59–5.35) |
| 2 | 6.00 (5.59–6.39) | 1.64 (1.52–1.77) | 4.35 (4.00–4.70) |
| 3 | 6.62 (6.21–7.04) | 1.54 (1.43–1.65) | 5.08 (4.71–5.45) |
| 4 | 7.60 (7.16–8.03) | 2.18 (2.04–2.32) | 5.41 (5.03–5.79) |
| 5 | 5.04 (4.72–5.87) | 1.41 (1.30–1.53) | 3.63 (3.35–3.90) |
| 6 | 4.37 (4.07–4.67) | 1.35 (1.22–1.48) | 3.02 (2.78–3.26) |
| 7 | 5.50 (5.12–5.87) | 1.65 (1.50–1.81) | 3.84 (3.52–4.16) |
| 8 | 4.94 (4.62–5.26) | 1.69(1.54–1.84) | 3.25 (2.99–3.51) |
| 9 | 4.10 (3.84–4.36) | 1.23 (1.13–1.33) | 2.87 (2.65–3.08) |
| 10 | 5.89 (5.52–6.26) | 1.79 (1.61–1.96) | 4.10 (3.80–4.40) |
| 11 | 5.92 (5.53–6.32) | 1.79 (1.65–1.93) | 4.13 (3.81–4.45) |
| 12 | 5.69 (5.31–6.06) | 1.31 (1.26–1.41) | 4.37 (4.04–4.70) |
| 13 | 5.29 (4.90–5.63) | 1.65 (1.48–1.82) | 3.62 (3.33–3.91) |
| 14 | 3.91 (3.65–4.17) | 1.23 (1.11–1.35) | 2.68 (2.47–2.89) |
| 15 | 3.52 (3.30–3.74) | 1.10 (1.01–1.19) | 2.41 (2.23–2.60) |
| 16 | 4.39 (4.14–4.64) | 1.34 (1.24–1.45) | 3.05 (2.84–3.26) |
| **Cumulative** | **5.35 (5.01–5.70)** | **1.56 (1.43–1.69)** | **3.08 (3.51–4.09)** |

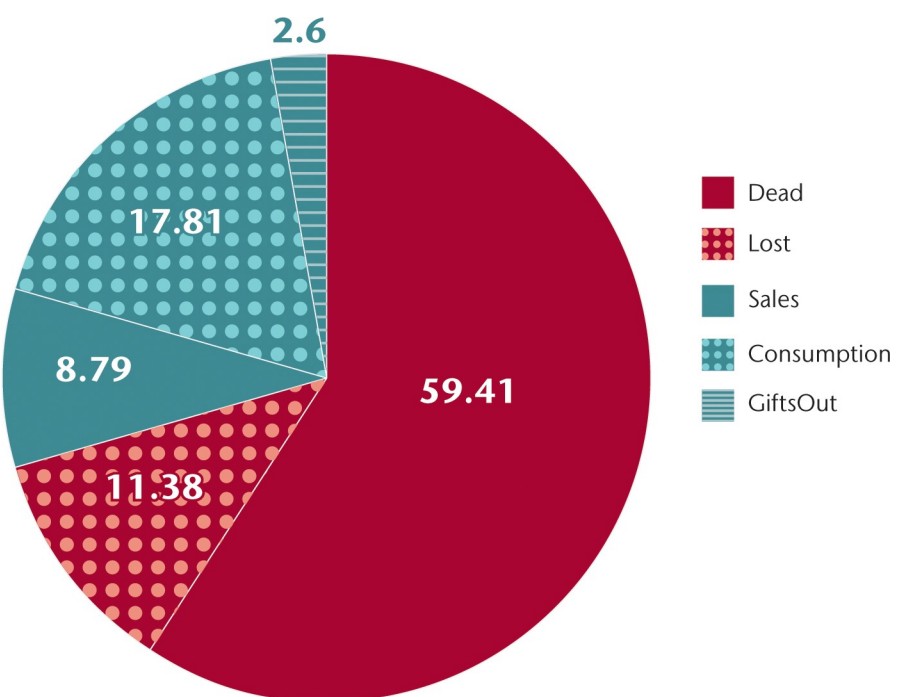

**Fig 2. Source of reported off-take in chickens (percentages).**

experiences, has been investigated in epidemiologic and medical research [18], where it has found to be related to factors including length of recall period, frequency of experiencing reported events, and respondent characteristics such as age. While retrospective surveys are often used in agricultural and livestock surveys, recall bias is rarely quantified because of the expense [19]. This study design is well placed to identify recall bias because it includes a respondent recall of chicken gains and losses as well as a longitudinal census. In this study, respondents are systematically over-reporting gains relative to losses of chickens. This may reflect a bias towards matching reported gains to the number of chickens at the time of the census and/or a tendency to discount prior, especially temporally distant, losses. The age of the chicken may also be a factor in recall bias. In scavenge-based and free-range systems, an estimated 70 percent of chicks die before they reach the age of six weeks [3]. While we do not have data on the proportions of chick losses relative to older chickens, one explanation is that prior losses of chicks may be discounted relative to adult chickens and thus lead to recall bias on losses. Another potential explanation is social desirability bias, whereby the desire to present oneself in a positive light or to give a socially desirable response can lead to systematic under-reporting of negative outcomes [20].

## Involuntary loss and opportunity cost

The hypothesis that chicken numbers are maintained at a low, relatively constant level due to a decision to maximize value while controlling opportunity cost linked to chicken management can be rejected. Both sources of involuntary loss, observed mortality and loss/predation, reflect, at least in part, lack of investment in chicken husbandry. Observed mortality is due to infectious disease, poor nutrition available by scavenging, and frequently a combination of the two [4,14,21]. Loss/predation reflects the lack of secure housing, especially during the day

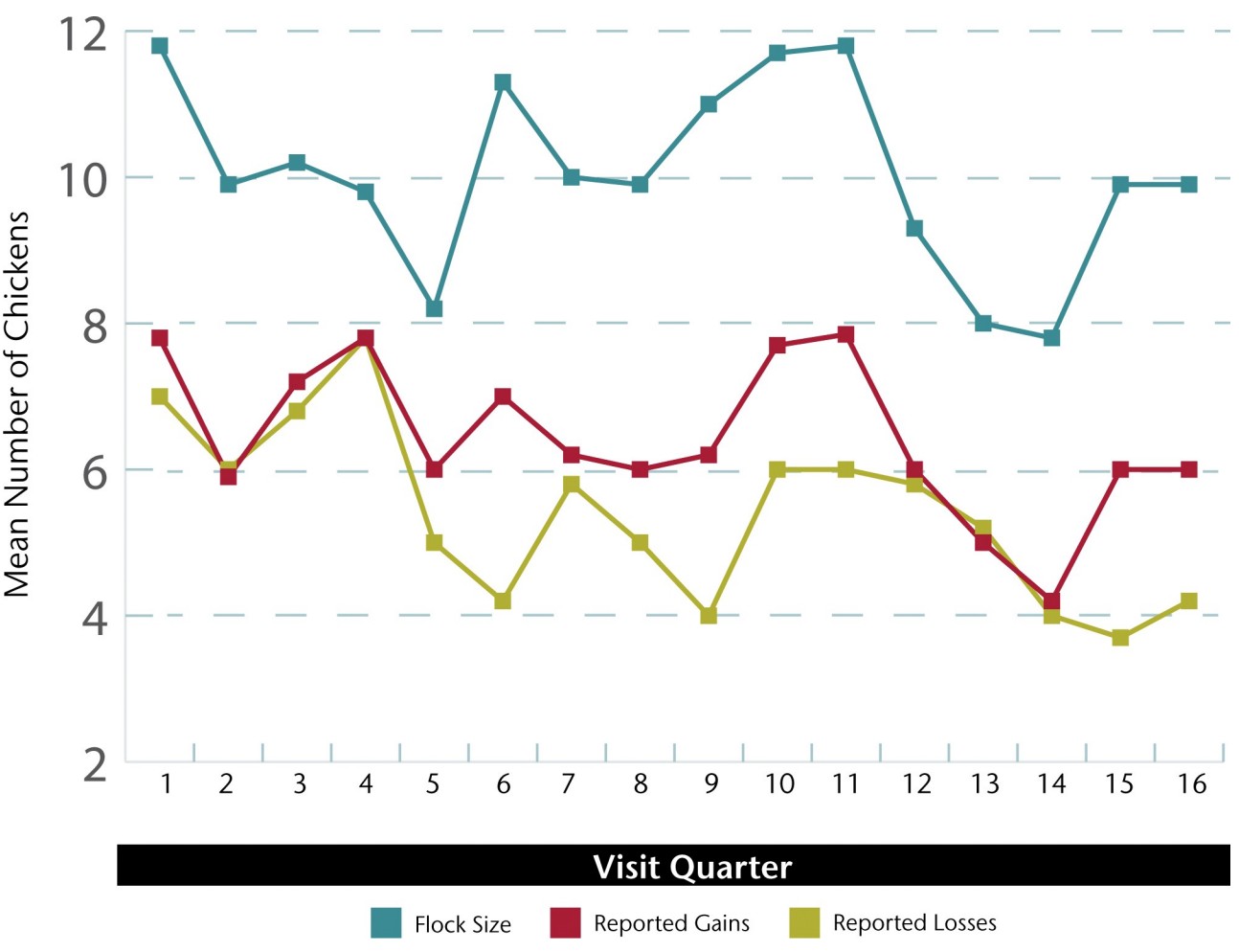

**Fig 3. Mean household flock size, reported gains and losses.**

when most chickens are freely scavenging. In our study, 98% of households reported that chickens free range scavenged during the day; 93% maintained chickens within the human household at night with only 7% in a dedicated coop or enclosure. Similarly, only 12% of households reported receiving some form of treatment, vaccination, or de-worming at some point during the 16 quarterly visits. This is supported by a survey of a subset of 533 households in which 5.65% reported veterinary service with vaccination at 1.35%, ectoparasite control at 1.26%, and de-worming at 0.42%.This lack of investment may reflect lack of capital available for vaccine and drug purchases and materials for chicken housing, the opportunity cost given other household responsibilities, or both. Accessible capital is limited within the households: mean assets for these households were reported in 2015 as $100 USD (median of $50 USD/ household) [15] and in the current study mean daily income was $1.17 USD (S1 Table). Similarly, time for chicken management may be markedly limited as 91% of households collect firewood daily for cooking and 60% of households devote more than an hour to daily water collection. Vaccination against Newcastle disease, an episodic and highly virulent disease responsible for the greatest mortality in small flocks [22,23], requires both capital and time [24,25]. Consequently, vaccinations and treatment for Newcastle disease and other poultry diseases may be difficult for smallholder households to routinely access [26].

### Maximizing value of smallholder poultry production

Chickens are the most commonly held livestock species in this study population [15], broadly consistent with their role in rural smallholder communities [2,27]. As a commonly held asset, there is potential to enhance their value to the household. Over 800 children were tracked for a research project that overlapped temporally and spatially with the current study [16]. Recent egg consumption was associated with an average 30% increase in children's height gain, and egg consumption, unsurprisingly, was linked to the number of chickens owned by the household [16]. In addition to direct nutritional benefits, the mean selling price for a chicken in the current study was $1.97 USD ($1.57–2.39 USD) (S2 Table), representing a potential gain for increasing voluntary off-take even if the same total number of chickens was maintained. Given a mean daily income of $1.17 USD, this represents an opportunity based on a currently widely held resource. To the degree that our study population is representative of rural African households in poverty, our findings suggest that initiatives solely focused on providing chickens or on improved breeds of chickens are unlikely to be successful in the absence of improved husbandry. While selected breeds have increased productivity in more intensively managed production, "local African chickens" are of composed of ecotypes with high genetic diversity and heterogenicity, a result of natural selection within the resource limitations of smallholder farmers [28]. As mortality represents the greatest source of involuntary loss, programs to reduce the capital and opportunity costs of vaccination and supplemental feed for the local chicken ecotypes are most likely to be effective in resource optimization. Comparative data from a recent study of smallholder chicken owners in Tanzania indicates that a substantial majority (81%) are aware of Newcastle Disease vaccines, even though only 26% had used these in the past three months [24]. A willingness to pay analysis from the same study population indicated that households were prepared to pay twice the actual cost of vaccine, reflecting recognition of the benefits of vaccination [11]. However, the opportunity cost to acquire vaccines and the need to share vaccine among multiple households to reduce the cost per dose represent barriers to routine vaccination [11,25]. Vaccine sharing, requiring leadership and coordination within the community, is enhanced through education; if the household decision-maker for chickens had completed primary school or secondary school compared to having no formal education, the household was 2.7 and 4.4 times, respectively, more likely to share vaccine [25]. Given that in the present study communities, 14% had no formal education and 65% had only primary school education (Table 1), this suggests a barrier to routine vaccination that could be overcome programmatically. While we do not have similar data on feed supplementation, similar capital limitations and opportunity cost barriers likely exist for smallholder households.

## Conclusion

This study strongly suggests that there is substantial opportunity to enhance the value of chickens as an asset, both nutritional and income generating, for smallholder households living at poverty level. Our findings suggest that programs emphasizing community level vaccination and feed supplementation are much more likely to be effective than those solely focused on providing chickens

## Supporting information

**S1 Table. Summary of quarterly household incomes (USD).**
(DOCX)

**S2 Table. Summary of quarterly income and on-farm expenses (USD).**
(DOCX)

## Acknowledgments

We thank the contributions of the following individuals, who made the research possible. Field team supervisor, James Oigo; Animal Health Technicians, Samwel Asembo, Geoffrey Odima, Bob Miseda, Beryl Oyoo and Austin Ochung'; Community Health Interviewers: Joseph Onyango, Daniel Odongo, Rosemary Warinda, Bridon Kojo, Thomas Wachiaya, Fredrick Ong'wen, Meresia Owuor; Martin Aundi, Fredrick Adhiambo, Benedine Warinda, Jane Okal, Fred Ojode; Data Programmer and Manager, Linus Otieno; Data Clerk, Judith Oduol.

## Author Contributions

**Conceptualization:** Guy H. Palmer.

**Data curation:** Zoë A. Campbell, Samuel M. Thumbi.

**Formal analysis:** Elkanah Otiang, Samuel M. Thumbi, Guy H. Palmer.

**Funding acquisition:** Guy H. Palmer.

**Investigation:** Elkanah Otiang, Samuel M. Thumbi.

**Methodology:** Elkanah Otiang, Zoë A. Campbell, Samuel M. Thumbi, Guy H. Palmer.

**Project administration:** Guy H. Palmer.

**Resources:** Guy H. Palmer.

**Supervision:** Samuel M. Thumbi, Lucy W. Njagi, Philip N. Nyaga, Guy H. Palmer.

**Validation:** Elkanah Otiang, Guy H. Palmer.

**Writing – original draft:** Elkanah Otiang, Guy H. Palmer.

**Writing – review & editing:** Elkanah Otiang, Zoë A. Campbell, Samuel M. Thumbi, Lucy W. Njagi, Philip N. Nyaga, Guy H. Palmer.

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
