## [Decision Letter · Decision Letter 0]

23 Apr 2020

PONE-D-20-06528

Mortality as the primary constraint to enhancing nutritional and financial gains from poultry: A multi-year longitudinal study of smallholder farmers in western Kenya

PLOS ONE

Dear Dr. Palmer,

Thank you for submitting your manuscript to PLOS ONE. After careful consideration, we feel that it has merit but does not fully meet PLOS ONE’s publication criteria as it currently stands. Therefore, we invite you to submit a revised version of the manuscript that addresses the points raised during the review process.

The manuscript was reviewed by three experts and while all three reviewers are very favorable, few very minor suggestions were made for the overall improvement.  Kindly respond to the critique when revising the manuscript.  Please include a response summary as well.  Thank you.

We would appreciate receiving your revised manuscript by Jun 07 2020 11:59PM. To enhance the reproducibility of your results, we recommend that if applicable you deposit your laboratory protocols in protocols.io, where a protocol can be assigned its own identifier (DOI) such that it can be cited independently in the future. For instructions see: http://journals.plos.org/plosone/s/submission-guidelines#loc-laboratory-protocols

We look forward to receiving your revised manuscript.

Kind regards,

Roman R. Ganta

Academic Editor

PLOS ONE

2. Please upload a copy of Supporting Information Table S1, S2 which you refer to in your text on page 11-12.

Reviewers' comments:

Reviewer's Responses to Questions

**Comments to the Author**

1. Is the manuscript technically sound, and do the data support the conclusions?

Reviewer #1: Yes

Reviewer #2: Yes

Reviewer #3: Yes

2. Has the statistical analysis been performed appropriately and rigorously? 

Reviewer #1: Yes

Reviewer #2: N/A

Reviewer #3: No

3. Have the authors made all data underlying the findings in their manuscript fully available?

Reviewer #1: Yes

Reviewer #2: Yes

Reviewer #3: Yes

4. Is the manuscript presented in an intelligible fashion and written in standard English?

Reviewer #1: Yes

Reviewer #2: Yes

Reviewer #3: Yes

5. Review Comments to the Author

Reviewer #1: This is a well conducted study and the data provided support the conclusion. Importantly, the manuscript provides empirical data needed to address poverty reduction by disease and nutrition management to improve poultry keeping, especially by women. The manuscript is well written.

Reviewer #2: Although this manuscript is descriptive, it offers baseline information that might for the basis for improvement of poultry production by small producers in Kenya. The manuscript is concise and well written. I suggest that, instead of numbering visits (1-16), could this be reported per season, say rainy and dry season for example? I also would like to see some concrete discussion on how these data could be utilized to improve poultry production. Please also comment on chicken breeds in villages and type of housing for chickens, also comment on veterinary service (available of nor available)

Reviewer #3: The article is well written, and its contents are valuable.

It provides evidence of significant constrains to nutritional and financial gains from poultry production system in rural communities in many ways in Africa.

6. PLOS authors have the option to publish the peer review history of their article (what does this mean?). If published, this will include your full peer review and any attached files.

Reviewer #1: No

Reviewer #2: No

Reviewer #3: No

---

## [Author Response · Author response to Decision Letter 0]

8 May 2020

We appreciate the reviewers’ time and effort in the critique of the manuscript and have responded to the comments below and note the specific changes in the revised manuscript.

Reviewer 1: The reviewer did not have any specific comments for change or improvement. We appreciate the comments.

Reviewer 2: The reviewer had 3 specific suggestions, each of which is addressed below and with changes to the manuscript.

2.1 “I suggest that, instead of numbering visits (1-16), could this be reported per season, say rainy and dry season for example?” All visits cannot be assigned to a given rainy/dry season as the actual start, duration, and intensity of the rainy seasons vary by year and many visits could not be confidently placed within a rainy/dry season. However, we identified a subset of 50 households for which visits could be placed by year into either a dry season or wet season. We then analyzed the same variables (gains/losses, source of gains/losses, sale price) by season. Based on this subset of households, there was no difference in any of the variable by season. We have added this finding to the manuscript.

2.2 “I also would like to see some concrete discussion on how these data could be utilized to improve poultry production.” We have modified the Discussion to emphasize the importance of preventable mortality through improved access to veterinary services. As the reviewer requests in the 3rd comment, we report the minimal use of veterinary services and integrate this finding with the barriers to smallholder poultry vaccination as recently reported by Campbell et al. (references 11,24).

2.3 “Please also comment on chicken breeds in villages and type of housing for chickens, also comment on veterinary service (available of nor available).” Defined as “local African chicken”, these chickens are described as “ecotypes” rather than “breeds” as there is no defined strategy for maintaining genetic lines. Across ecotypes there is marked diversity, significantly greater than in defined poultry breeds. We have added this to the manuscript and provided a contemporary reference (28). We have also provided an analysis of housing: 98% of households allowed chickens to scavenge during the day; 93% maintained chickens within the human household at night with 7% in a dedicated coop or enclosure. This has been added to both the Results/Discussion. We also addressed use of any veterinary service: 12% of households reported receiving some form of treatment, vaccination, or de-worming during the survey period. To this data, we have added in the Results/Discussion reference to a second set of more complete data on a subset of 533 households in which 5.65% reported veterinary service with vaccination at 1.35%, ectoparasite control at 1.26, and de-worming at 0.42%. 

Reviewer 3: The reviewer had 2 specific suggestions, each of which is addressed below and with changes to the manuscript. In addition, the reviewer recommended several minor grammatical edits to the manuscript.

3.1 “The title page and the manuscript body does not conform to PLos One guideline and should be formatted as such. Also, there should be page and line numbering in the entire manuscript.” We have revised the manuscript to be in compliance with PLoS One guidelines and have included page and line numbering.

3.2 “As stated by the authors, in spite of the large population of village chickens in rural communities in Africa, their productivity is low due to various factors including high mortality rates, especially in chicks. While village chickens do not only play an important role in the supply of protein and income to resource-poor households. It is important to recognize the relevance of rural chickens in occasions such as religious and cultural events in the African traditional settings.” We appreciate this comment as some of this significance is reflected in gifts and consumption as sources of both gain and loss of chickens at the household level. We have added this important role, with an accompanying relevant reference, to the Introduction. 

3.3 “Other minor review comments are included in the attached manuscript for authors consideration.” We have made the suggested grammatical changes.

---

## [Editor Report · Decision Letter 1]

12 May 2020

Mortality as the primary constraint to enhancing nutritional and financial gains from poultry: A multi-year longitudinal study of smallholder farmers in western Kenya

PONE-D-20-06528R1

Dear Dr. Palmer,

We are pleased to inform you that your manuscript has been judged scientifically suitable for publication and will be formally accepted for publication once it complies with all outstanding technical requirements.

With kind regards,

Roman R. Ganta

Academic Editor

PLOS ONE
---

## [Editor Report · Acceptance letter]

15 May 2020

PONE-D-20-06528R1 

Mortality as the primary constraint to enhancing nutritional and financial gains from poultry: A multi-year longitudinal study of smallholder farmers in western Kenya 

Dear Dr. Palmer:

I am pleased to inform you that your manuscript has been deemed suitable for publication in PLOS ONE. Congratulations! Your manuscript is now with our production department. 

With kind regards,

on behalf of

Dr. Roman R. Ganta 

Academic Editor

PLOS ONE